# On the Thermal Capacity of Solids

**DOI:** 10.3390/e24040479

**Published:** 2022-03-29

**Authors:** Armin Feldhoff

**Affiliations:** Institute of Physical Chemistry and Electrochemistry, Leibniz University Hannover, Callinstraße 3A, D-30167 Hannover, Germany; armin.feldhoff@pci.uni-hannover.de; Tel.: +49-511-762-2940

**Keywords:** heat capacity, entropy capacity, susceptibility, Debye model, Sommerfeld coefficient, graphite, diamond, barium titanate, phase transition, reaction entropy

## Abstract

The term thermal capacity appears to suggest a storable thermal quantity. However, this claim is not redeemed when thermal capacity is projected onto “heat”, which, like all energy forms, exits only in transit and is not a part of internal energy. The storable thermal quantity is entropy, and entropy capacity is a well-defined physical coefficient which has the advantage of being a susceptibility. The inverse of the entropy capacity relates the response of the system (change of temperature) to a stimulus (change of entropy) such as the fluid level responses to a change in amount of fluid contained in a vessel. Frequently, entropy capacity has been used implicitly, which is clarified in examples of the low-temperature analysis of phononic and electronic contributions to the thermal capacity of solids. Generally, entropy capacity is used in the estimation of the entropy of a solid. Implicitly, the thermoelectric figure of merit refers to entropy capacity. The advantage of the explicit use of entropy capacity comes with a descriptive fundamental understanding of the thermal behaviour of solids, which is made clear by the examples of the Debye model of phonons in solids, the latest thermochemical modelling of carbon allotropes (diamond and graphite) and not least caloric materials. An electrocaloric cycle of barium titanate close to its paraelectric–ferroelectric phase transition is analysed by means of entropy capacity. Entropy capacity is a key to intuitively understanding thermal processes.

## 1. Introduction

### 1.1. Energy and Entropy

In the traditional approach of thermodynamics, which identifies “heat” as thermal energy [1], “heat capacity” is a inadequate term. The word *capacity* conveys the notion that the quantity “heat” is contained by the receiving vessel, however, it has been pointed out by several authors that “heat” cannot be stored in a system (e.g., solid). In his paragraph on the concept of “heat”, Zemansky [2] (p. 76) wrote that “‘heat’ is internal energy in transit” and “it would be incorrect to refer to the ‘heat’ in a body”. Similarly, Callen [3] (p. 112) wrote that “‘heat’ refers to a mode of energy flux rather than to an attribute of a state of a thermodynamic system”. Strunk [4] directly explicated that “‘heat’ is the strange thing that is flowing only, but disappears upon arrival in any system”. Falk and Ruppel [5] (p. 92) emphasised that “‘heat’ is not a part of internal energy, but an energy form. The fact that ‘heat’ is not in a system but only occurs when energy is exchanged, like all energy forms [6], is one of the most crucial points in thermodynamics, which cannot be stated often enough [7]”.

By principle, a “heat” current is coupled to an entropy current [5] (p. 92). As accentuated by Falk et al. [8] “...one must focus on the substance-like quantities accompanying the flow of energy if one wants to get a suitable description of energy transfer.” It is helpful to consider entropy as an energy carrier [8]. The amount of energy carried by entropy is “heat”. It only makes sense to speak about “heat” (thermal energy) when entropy flows. Interestingly, Callen [3] (p. 32) wrote that “a quasi-static flux of ‘heat’ into a system is associated with an increase of entropy of that system” [9]. Because there is no such thing as stored thermal energy, but stored entropy, it is reasonable to consider the entropy capacity of a solid. Although little known, entropy capacity is a well-defined coefficient carrying the real meaning of a capacity and can be used in that manner with great advantage to the understanding of thermal processes and has been addressed by several authors.

### 1.2. Outline

The aim of this work was to provide access to sources of the dispersed knowledge on entropy capacity and to illustrate the usefulness of this concept. After an overview of the existing literature on this topic, basic relationships are reviewed. Then, the part of Wiberg’s textbook [10] related to entropy capacity is recapitulated, which gives a vivid picture of entropy capacity in general and of carbon allotropes in particular. A bridge is built between fundamental considerations and current fields of application by presenting the example of caloric materials and thermoelectrics. The phonon-related entropy capacity of solids is discussed with respect to the Debye model by examples given in Debye’s classical work [11]. In addition, the concept of entropy capacity is expanded to the electronic contribution using the model of the free electron gas at low temperature. Entropy capacity is often implicitly used when separating electronic and phononic contributions from the “heat capacity”. The discussion turns to persistent confusion due to the disruptive development of thermodynamics; a probable resolution is to leave dead metaphors behind.

## 2. Materials and Methods

Experiments to illustrate the analogy with a fluid vessel were performed using red wine—specifically Monopoles Nicola Napoléon Bordeaux Superior 2018 (Nicola Napoléon CIE & S.A.R.L, Saint-Émilion, Gironde, France, packager code EMB 33394)—and a glass of the type Schott Zwiesel Whisky Nose 120 (Zwiesel Kristallglas AG, Zwiesel, Germany). Video recording and photographing were performed using a Sony DSC-RX100 Mark 3 digital camera (Sony Corporation, Tokyo, Japan). Items were placed on a portable shooting table (Calumet Photographic Inc., Chicago, IL, USA) and the scene was illuminated using two Nanlite Lumipad 25 (Guangdong Naguang Photo & Video Systems Co., Ltd., Shantou City, Guangdong, China). Video editing was performed using HitFilm Express 14 (FXhome, Norwich, Norfolk, UK). The music in the videos is “Cute” from Bensound.com. Photo editing was performed using Image J, version 1.53o (Wayne Rasband, US National Institutes of Health, Bethesda, MD, USA).

Calculations of the graphs were performed using Python embedded into OriginPro, Version 2022 (OriginLab Corporation, Northampton, MA, USA). Graphs were set and analysed in OriginPro. Composite figures were arranged in PowerPoint in Microsoft Office Professional Plus 2016 (Redmond, WA, USA) and exported in portable data format (PDF).

## 3. Entropy Capacity

Lunn [12] (p. 2) stated that “at constant volume, the capacity of an ideal gas for change of thermal energy is constant but its capacity for change of entropy varies inversely as the absolute temperature”, which refers to the Dulong–Petit relation of the ideal gas.

Falk [13,14] recalled that entropy capacity (at the time known as heat capacity) was introduced by Joseph Black (1728–1799), who refined the term heat (caloric) that has been around for centuries. Entropy is a resurrection of the caloric [13,15,16,17] and endows entropy capacity with the real meaning of capacity. Falk made clear that entropy capacity is a susceptibility, i.e., a second derivative of a Massieu–Gibbs function with respect to intensive variable(s), and must be positive within the stability boundaries of the system. The inverse of the entropy capacity, which has been called *heatability* by Herrmann and Hauptmann [18] (p. 28ff), relates the response of the system (change of temperature) to a stimulus (change of entropy contained). The metrology of entropy capacity is addressed and so is the fact that at least two different entropy capacities need to be considered for a gas, e.g., at constant volume and at constant pressure, because the entropy contained in gas depends not only on temperature, but also noticeably on pressure. Falk and Ruppel [5] (p. 297f) addressed these aspects in a condensed form.

Strunk [19] (pp. 57f, 331) mentioned that (specific) entropy capacities are susceptibilities and can be obtained either by differentiating the entropy of the system with respect to the temperature or by dividing the “heat capacity” by the absolute temperature. He used the latter relationship to formulate susceptibility matrices for simple phases, which are symmetric because of the Maxwell relations, and each comprises an entropy capacity on its diagonal, respectively, [20] (pp. 7, 11). Constraints explicitly considered with respect to entropy capacities are the intensive variables pressure and chemical potential being constant.

Mareŝ et al. [21] criticised the inconvenient choice of the conceptual basis of thermodynamics created in the 19th century and identified entropy with the caloric (heat) of older theories. Entropy capacity [22] is presented by differentiating the entropy of the system with respect to temperature.

Job [23,24] treated several aspects of entropy capacity and explicitly addressed the Debye model of solids (p. 114f). The change from parabolic dependence (Debye model) to hyperbolic dependence (Dulong–Petit relationship) on temperature is interpreted as a phase transition, which is characterised by a maximum entropy capacity at approximately a quarter of the Debye temperature. In their undergraduate textbook, Job and Rüffler [25] dealt with molar and specific (per mass) entropy capacities in analogy to matter capacity and buffer capacity with demonstrative examples. The accompanying workbook [26] comprises several exercises on entropy capacity and provides the corresponding detailed solutions with helpful comments.

Fuchs [27,28,29,30] has developed the most extensive view on entropy capacity to date, which uses analogies to gravitation, hydraulics, electricity, and mechanics, (i.e., mass as momentum capacitance). With respect to heating, he showed that entropy capacity relates the rate of change of temperature to the rate of change of the entropy. He mentioned that the direct measurement of entropy capacity is not simple and addressed the difficulties involved. He presented a temperature–entropy capacity diagram for the ideal gas and tables for the entropy capacity of some substances. A formulaic expression for the entropy capacity of phonons, according to the low-temperature approximation of the Debye model, is given. The entropy capacity of black body radiation is treated. Consider the entropy capacity at the constant magnetisation of a paramagnetic substance leads to a vivid and simple interpretation of magnetocalorics. Thoroughly analysed examples as well as detailed explanations and exercises with solutions [28] are provided. Fuchs et al. [31,32] put entropy capacity into the context of the historical development of the caloric theory and linked its value under different constraints (i.e., constant volume or constant pressure) via the adiabatic coefficient.

In his student textbook, Wiberg [10] vividly demonstrated that the chemical substances are capacities for entropy. The abstract terms entropy and reaction entropy are substantiated as capacity factors for thermal energy analogous to charge in electricity and the amount of fluid (water) in hydraulics. The intensity factor is then the absolute temperature analogously to electrical potential in electricity and the height of fluid level in hydraulics. Wiberg [10] (p. 140) wrote: “When ‘heat’ is supplied to a chemical substance, its entropy content is increased. In the same way, the fluid level (i.e., the height of the amount of water) is raised while filling a water vessel with water, the entropy level (i.e., the temperature of the respective chemical substance) is raised while filling an entropy vessel (e.g., a gas or a liquid [or a solid]) with entropy. In both cases, the increase in height is dependent on the shape of the vessel [33].” The shape of the entropy vessel is given by the entropy capacity of the chemical substance. In the example of the allotropic phase transition from graphite to diamond or vice versa, which he discussed in a general concept of chemical reactions, Wiberg very clearly showed the consequence of the changed shape of the entropy vessel, which can cause the emission of entropy or absorption of entropy. Analogous to traditional adjectives exothermic and endothermic reactions, the adjectives exotropic and endotropic are suggested, which allow distinguishing reactions with entropy being released (negative reaction entropy) from reactions with entropy being absorbed (positive reaction entropy), because of increased or decreased entropy capacity of products compared to educts. Wiberg gives very detailed figures for the amount of entropy stored in graphite and diamond at different temperatures.

## 4. Entropy Capacity versus “Heat Capacity”

The entropy capacity *K* relates the change in entropy *S* with the absolute temperature *T*:(1)$$K:=\frac{\partial S}{\partial T}$$
regardless of the constraints [13] of constant volume *V* and a constant number of particles *N*:(2)$$K_{V,N}={\left(\frac{\partial S}{\partial T}\right)}_{V,N}$$
or constant pressure *p* and a constant number of particles *N*:(3)$$K_{p,N}={\left(\frac{\partial S}{\partial T}\right)}_{p,N}$$

If the number of particles is implicitly kept constant, these quantities can be denoted as the entropy capacity at constant volume $K_{V}$ or the entropy capacity at constant pressure $K_{p}$.

The “heat capacity” $C_{V}$ is related to the entropy capacity $K_{V}$ at constant volume by Equation (Equation 4), which refers to the change in internal energy *E* and thus transferred energy. Following the approach of Fuchs [29], it is semantically more appropriate to call $C_{V}$ the temperature coefficient of energy, which reflects its real meaning:(4)$$C_{V}=T\cdot K_{V}=T\cdot {\left(\frac{\partial S}{\partial T}\right)}_{V}={\left(\frac{\partial E}{\partial T}\right)}_{V}$$

The “heat capacity” $C_{p}$ is related to the entropy capacity $K_{p}$ at constant pressure by Equation (Equation 5), which refers to the change in enthalpy *H* and thus the transferred enthalpy. Following the approach of Fuchs [29], it is semantically more appropriate to call $C_{p}$ the temperature coefficient of enthalpy, which reflects its real meaning:(5)$$C_{p}=T\cdot K_{p}=T\cdot {\left(\frac{\partial S}{\partial T}\right)}_{p}={\left(\frac{\partial H}{\partial T}\right)}_{p}$$

The fact that $C_{V}$ and $C_{p}$ refer to the exchange of different quantities was discussed by Falk [13] (p. 188). Falk stated that the term “heat capacity” is linguistically and conceptually a trap.

Zemansky [2] (p. 306) stated that the expression derived from the partition function in statistical thermodynamics for entropy is simpler than the expression derived for the internal energy. Interestingly, when Callen [3] (p. 353f) treated the Debye model of solids, he did not derive the contribution of the phonons to the internal energy but to the molar entropy. As such, he implicitly used the entropy capacity to deduce the “heat capacity” as was performed herein in Equations (Equation 4) and (Equation 5). If temperature is known in addition to the values of one or the other, $C_{V}$ and $K_{V}$ or $C_{p}$ and $K_{p}$ are easily convertible. Values of entropy capacity for some substances are given in [25,29].

## 5. Analogy: Storage of a Fluid in a Vessel

Wiberg [10] drew an analogy between the capacity of chemical substances to store entropy and the hydraulic capacity of a vessel to store a fluid. The latter is illustrated in Figure 1. The capacity of a glass to store a fluid depends upon its shape, which perhaps changes with the fluid level. In Figure 1, the fluid level is subsequently raised by equal height differences of 25 mm each, but the respective amount of fluid is quite different in each step, because of the shape of the vessel being wider or narrower.

Of course, the chemical substances are containers for entropy with permeable walls. Due to entropy permeation through nonadiabatic walls, at a certain rate, the entropy level in the container will drop when the temperature (entropy level) in the surrounding decreases, and the entropy level will rise when the temperature in the surrounding increases. For solids, equilibration takes a long time [34]. The situation with chemical substances in general is even more than intricate because the flow of entropy under nonisothermal conditions is associated with the production of additional entropy [23,25,29]. Nevertheless, for storing entropy in chemical substances, the analogy is very instructive.

## 6. Entropy Capacity of Diamond and Graphite

The shape of the entropy vessel in Figure 2a corresponds to graphite and that in Figure 2b to diamond under isobaric conditions. The hatched area in Figure 2a marks an infinitesimal amount of entropy $dS=K_{p}\cdot dT$, which is linked to an infinitesimal temperature interval dT by the entropy capacity $K_{p}$. The wider the vessel is, the more entropy dS must be filled in to increase the entropy level (i.e., the temperature) by dT. Easily, the entropy *S* contained in the vessel at a certain entropy level (i.e., temperature *T*) can be estimated according to Equation (Equation 6):(6)$$S\left(T\right)=\int_{0}^{T}K_{p}\cdot dT\;$$

Interestingly, the entropy is commonly estimated by such integrals with the integrand being $C_{p}\;/\;T$, which implicitly refers to the isobaric entropy capacity (as can be seen in Equation (Equation 5)). Using the entropy capacity explicitly comes with the benefit of clarity. The entropy stored at equivalent temperature intervals of 300 K each can be estimated the same way and is added to these intervals in Figure 2a for graphite and in Figure 2b for diamond. From Figure 2c, diamond is obviously the narrower entropy vessel compared to graphite. The amount of entropy being stored less in diamond compared to graphite in each 300 K interval is added to the graph. The accumulated entropies of the present paper are in qualitative agreement with Wiberg’s book [10] (see Table A2), except that Wiberg equated, perhaps for didactic reasons, the entropy capacities of both carbon allotropes from 600 K upward.

The graphs *T* as a function of $K_{p}$ in Figure 2 show the inverse of the entropy capacity, i.e., ∂T/∂S, which has been called heatability by Herrmann and Hauptmann [18] (p. 28ff).

Usually, the diagrams are plotted as in Figure 3 with the temperature in the axis of ordinates and the entropy capacity on the axis of abscissae, which, however, does not change anything regarding its vivid meaning [10]. In Figure 3, in addition to the molar isobaric entropy capacity ${\hat{K}}_{p}$ of graphite and diamond, the molar isochoric entropy capacity ${\hat{K}}_{V}$ of the Dulong–Petit relationship (hyperbolic) is plotted. The respective ${\hat{C}}_{p}$ and ${\hat{C}}_{V}$ curves are given in Figure A1, but lack vivid meaning. Above 1800 K, ${\hat{K}}_{p}$ of graphite converges to the classical Dulong–Petit relationship, which here is $3\cdot R\cdot T^{-1}$ [36] (p. 427), and eventually exceeds it, while ${\hat{K}}_{p}$ for diamond remains below. Here, *R* is the universal gas constant.

Graphite is an extreme example that cannot be described by a Debye model with a single Debye temperature. Its phonon density of states is likely to be excessively complex, which is due to its strong anisotropy with weak van der Waals interplane forces and strong covalent bonds in the basal plane [37]. Even though diamond does not have such anisotropy, the Debye model oversimplifies the phonon dispersion and gives an inappropriate prediction of the entropy capacity or the temperature coefficient of enthalpy [37]. In a heuristic approach, some researchers have introduced a temperature-dependent Debye temperature [38], which allows the maintenance of the Debye model at varying temperature ranges [39] (p. 105ff), but thwarts the intention to predict the temperature dependence of the isobaric entropy capacity or the temperature coefficient of enthalpy over a wide temperature range using a single parameter [36] (p. 459f). As a consequence, quite different values of the Debye temperature have been reported for diamond depending upon how the Debye model was fitted to low-, mid- or high-temperature empirical data. Examples are given in Figure A2.

Moreover, the difference between $K_{p}$ or $C_{p}$ (empirical) and $K_{V}$ or $C_{V}$ (model) can be significant at high temperatures exceeding 10% [40]. The temperature dependence of the thermal expansion, isothermal compression, and molar volume must be considered to match the models to empirical data at higher temperatures. Due to incomplete data at high temperature, the combination of the aforementioned parameters to the Grüneisen parameter is often considered. With the assumption of a temperature-independent Grüneisen parameter, which implies that the Debye temperature is only dependent on the molar volume, the ratio $K_{p}\;/\;K_{V}$ or $C_{p}\;/\;C_{V}$ can be estimated [39] (p. 102ff). The current approach to provide model data for the calculation of phase diagrams (Calphad) databases [37,41], however, is multiparameter fitting to empirical data. The model proposed by Bigdeli et al. [37] relies on multiple Einstein temperatures and gives reasonable estimates for a wide temperature range, but fades away from empirical data above 3000 K (graphite) or 1000 K (diamond). Recently, a reliable description of $C_{p}$ of the carbon allotropes diamond and graphite from 0.1 K to the melting point has been given by Vassiliev and Taldrik [35] using a Debye–Maier–Kelley hybrid model, and was used in this work with the parameters listed in Table A1 to describe isobaric entropy capacity.

## 7. Reaction Entropy

Wiberg [10] considered the allotropic phase transition of diamond to graphite as the special case of a chemical reaction. If the transformation of 1 mol diamond to 1 mol graphite is considered, integrating the difference between the entropy capacities of the product (i.e., graphite) and reagent (i.e., diamond) in Figure 3 leads to the molar reaction entropy $\Delta \hat{S}$ according to Equation (Equation 7):(7)$$\Delta {\hat{S}}_{\mathrm{diamond}\to \mathrm{graphite}}=\int_{0}^{T}\left({\hat{K}}_{p,\mathrm{graphite}}\left(T\right)-{\hat{K}}_{p,\mathrm{diamond}}\left(T\right)\right)\cdot dT\;$$

The molar reaction entropy according to Equation (Equation 7) is plotted in Figure 4 versus temperature.

The reaction entropy of this work is in qualitative agreement with Wiberg’s book [10] (see Table A2), except that Wiberg equated, perhaps for didactic reasons, the entropy capacity of both carbon allotropes from 600 K upward. Therefore, in Wiberg’s diagram, reaction entropy is constant from 600 K upward, but in Figure 4, the reaction entropy further increases at a decreasing rate. The reaction entropy at integral multiples of 300 K is indicated by horizontal dashed lines and corresponds to subsequent sums of the values in Figure 3.

In an inert atmosphere (absence of oxygen), diamond can be heated to approximately 2000 K. However, its surface is covered by a thin layer of graphite [35,42]. If the transformation is considered to proceed at constant temperature, the reaction entropy is isothermally absorbed. Otherwise, the temperature of the resulting graphite would temporarily drop until the reaction entropy balances the entropy level. Following the term endothermic, Wiberg [10] coined such a process with positive reaction entropy as *endotropic*. The reverse reaction, whereby graphite is transformed into diamond, has been reported to appear at high temperature (ca. 1573 K to 3573 K), which was achieved by flash heating and high pressure (ca. 15 GPa), which were applied to keep the graphite at a strictly constant volume [43,44]. Under this high pressure, due to a rigidly fixed volume, the entropy capacity was expected to be smaller than the entropy capacity given in Figure 2a. The latter implicitly refers to a constant ambient pressure (ca. 100 kPa). The transformation of graphite into diamond has a negative reaction entropy and can thus be classified as an *exotropic* reaction. The classification is given as follows:Endotropic reaction, ΔS>0: entropy is *isothermally absorbed* by the chemical substance(s) from the environment (Ref. [10], p. 155, Ref. [25], p. 231ff).Exotropic reaction, ΔS<0: entropy is *isothermally ejected* from the chemical substance(s) to the environment (Ref. [10], p. 155, Ref. [25], p. 231ff).

Illustrative examples of the isothermal squeezing out or soaking up of entropy (sponge model) are given in [23,25,45]. If the reaction cannot be exchanged with the environment at a sufficiently fast rate, however, the temperature of the chemical substance(s) temporarily changes. In the extreme case, the temperature change is adiabatic. This is discussed in the example of an electrocaloric material in Section 8.

Wiberg [10] integrated a graph analogous to Figure 4 to produce a vivid picture of the Gibbs–Helmholtz equation and deduced graphs of Gibbs free energy and enthalpy versus temperature for the transformation of diamond into graphite. Wiberg clearly deduced that endotropic reactions are possible at decreasing temperature only if the entropy to be absorbed (ΔS>0) is sufficiently large to compensate for the Gibbs free energy (e.g., transformation, evaporation, dissociation). In contrast, highly endothermic reactions with small reaction entropy only occur at very high temperatures. Exotropic reactions (ΔS<0) with large reaction entropy require highly exothermic conditions to occur at high temperature, while weakly exothermic reactions with small reaction entropy to be absorbed are only possible at low temperature (e.g., condensation, association). Thus, knowledge of reaction entropy as a function of temperature for the system of products and the system of reagents is important to obtain a vivid picture of possible reactions, and reaction entropy is closely linked to the entropy capacity of these systems.

In general, the molar reaction entropy $\Delta \hat{S}$ can be estimated according to Equation (Equation 8) by integrating the difference of molar entropy capacities of products ${\hat{K}}_{p,i}$ and reagents ${\hat{K}}_{p,j}$ weighted by respective stoichiometric coefficients $\nu_{i}$ and $\nu_{j}$:(8)$$\Delta \hat{S}\left(T\right)=\int_{0}^{T}\left(\sum_{i}^{\mathrm{products}}\nu_{i}\cdot {\hat{K}}_{p,i}\left(T\right)-\sum_{j}^{\mathrm{reagents}}\nu_{j}\cdot {\hat{K}}_{p,j}\left(T\right)\right)\cdot dT\;$$

## 8. Caloric Materials

So-called caloric materials often exhibit polymorphic phase transitions, which cause the absorption or release of entropy due to changed entropy capacity, and are triggered by magnetic stress (magnetocaloric [46]), mechanical stress (elastocaloric [47]), electrical field (electrocaloric [48,49]) or hydrostatic pressure (barocaloric [45]).

Using the example of magnetocaloric materials, Fuchs [29] (p. 234) discussed the coupling of magnetic and thermal processes. The flow of entropy from the environment into the material or out of the material into the environment depends on the latent entropy (with respect to magnetisation) and the entropy capacity $K_{M}$ at constant magnetisation *M*:(9)$$K_{M}={\left(\frac{\partial S}{\partial T}\right)}_{M}$$
directly leading to the latent entropy with respect to magnetisation (the extensive magnetic quantity). Latent entropy is related to the isothermal change of entropy [25] (p. 85) and it coincidences with what has been called reaction entropy in context of Equation (Equation 7) and Equation (Equation 8). “The term *latent* denotes the property of entropy *not* to affect the temperature of the system during phase change [29] (p. 191f).” The common understanding of entropy is that it changes the temperature of a system. When it does not, it is termed *latent* entropy in contrast to *sensible* entropy. Latent entropy (i.e., latent reaction entropy) gives an illustrative view of the isothermal absorption of entropy when the magnetisation is lowered. Upon lowering magnetisation, the entropy vessel becomes wider and can store more entropy at a given temperature. Recall that the relationship of stored entropy to the entropy capacity is given by Equation (Equation 6).

When considering the adiabatic demagnetisation of a paramagnetic substance, which is used to reach ultralow temperature, the entropy capacity $K_{H}$ at a constant magnetic field H (the intensive magnetic quantity) is used:(10)$$K_{H}={\left(\frac{\partial S}{\partial T}\right)}_{H}$$

These views can easily be extended to other members of the family of caloric materials with appropriate entropy capacity to be identified. Considering the multitude of thermal cycles that are possible, countless entropy capacities may be considered. Figure 5 provides some examples with either intensive or extensive fixed quantities. The respective symbols are explained in Table 1.

Giant electrocaloric effects have been reported for single-crystal BaTiO${}_{3}$ [50]. In refs. [48,51], the theoretical electrical entropy versus temperature diagram for BaTiO${}_{3}$ is discussed for different strengths of the applied electrical field. By differentiating these curves with respect to temperature, the electrical entropy capacity at constant electrical field $K_{ℇ}$ can be obtained. However, preference is usually given to the empirical data on the specific temperature coefficient of energy at the constant electrical field ${\tilde{C}}_{ℇ}$ (see Figure A3b), which were reported by Bai et al. [52]. These data were used to deduce the specific entropy capacity at a constant electrical field ${\tilde{K}}_{ℇ}$ for zero field and $ℇ=10\;\mathrm{kV}\cdot {\mathrm{cm}}^{-1}$ (see Figure A3b).

The electrocaloric cycle given by Scott [48] is adapted to barium titanate and analysed in Figure 6 with respect to entropy capacity. The cycle starts at zero field at a temperature of 412 K, slightly above the paraelectric–ferroelectric phase transition. With the electrical field applied in an adiabatic process, BaTiO${}_{3}$ becomes a narrower entropy vessel, which can store the initial entropy only with the entropy level (i.e., temperature) increased by ΔT=0.8K to 412.8 K. Then, the reaction entropy $\Delta S=1.45\;J\cdot K^{-2}\cdot {\mathrm{kg}}^{-1}$ is ejected and the temperature decreases to 412 K again. In another adiabatic process, the electrical field is decreased to zero again, which makes BaTiO${}_{3}$ a wider entropy vessel, and its temperature decreases to 411.2 K. Then, the entropy of an amount equal to the reaction entropy is absorbed, the temperature rises to 412 K, and the cycle is closed. The process is driven by polarisation energy and leads to the pumping of thermal energy. Note that arrows related to energy forms have different thicknesses. Reaction entropy ΔS is ejected at a higher temperature than that at which it is absorbed, which makes the associated thermal energy ejected in the warm leg of the cycle larger than the thermal energy absorbed in the cold leg of the cycle. The absorption or ejection of polarisation energy or thermal energy changes the internal energy, but none of the energy forms are part of the internal energy.

The figures given here for entropy and temperature are not superbly accurate because the graphs given in [52] were sampled in steps of only 0.5 K and interpolated to 0.1 K steps using an Akima spline fit. Bai et al. [52] reported a specific reaction entropy of $\Delta \tilde{S}=1.9\;J\cdot K^{-2}\cdot {\mathrm{kg}}^{-1}$ (ΔT=1.6K) at 412 K. The values in [52] were estimated from empirical ${\tilde{C}}_{ℇ}\left(T,ℇ\right)$ (see Equation (Equation 25)) and *T* values using the relationship equivalent to Equation (Equation 11), but without explicitly mentioning entropy capacity:(11)$$\Delta \tilde{S}=\int_{0}^{T}\left({\tilde{K}}_{ℇ}\left(T,ℇ\right)-{\tilde{K}}_{ℇ}\left(T,0\right)\right)\cdot dT\;$$

In Figure 6 and the discussion given above, irreversibility is omitted for clarity. For the treatment of *generated* reaction entropy in addition to *latent* reaction entropy, the reader is referred to [25] (p. 241ff).

## 9. Thermoelectrics and Thermal Conductivity

The thermoelectric figure of merit f=zT can be expressed as
(12)$$f=\frac{\mathrm{power}\;\mathrm{factor}}{\Lambda }=\frac{\mathrm{power}\;\mathrm{factor}}{\lambda }\cdot T≔zT$$
with the open-circuited specific thermal conductivity expressed either as entropy conductivity Λ or “heat” conductivity λ, which are related by the absolute temperature *T* according to λ=T·Λ [53,54]. An established method to measure the thermal conductivity is based on light flash analyser, which estimates the thermal diffusivity $D_{\mathrm{th}}$ [34]. When the density ρ and the specific isobaric “heat capacity” ${\tilde{C}}_{p}$ or the specific isobaric entropy capacity ${\tilde{K}}_{p}$ are also known, the “heat” conductivity:(13)$$\lambda =D_{\mathrm{th}}\cdot \rho \cdot {\tilde{C}}_{p}$$
or the entropy conductivity:(14)$$\Lambda =D_{\mathrm{th}}\cdot \rho \cdot {\tilde{K}}_{p}$$
can be obtained. The thermal diffusivity $D_{\mathrm{th}}$ can be regarded as the diffusion coefficient of “heat” as well as the diffusion coefficient of entropy [19]. With temperature *T* explicitly showing up in the right part of Equation (Equation 12), entropy conductivity is implicitly used (left part of Equation (Equation 12)), which implicitly refers to the entropy capacity according to Equation (Equation 14). Notes on Fourier’s original work support the view to centre considerations on thermal conductivity around a storable quantity [31].

## 10. Phononic Contributions to Entropy Capacity: Debye Model

According to Equations (Equation 23) and (Equation 22), the molar isochoric entropy capacity of the phonon gas with Debye temperature $\Theta_{D}$ is given by Equation (Equation 15):(15)$${\hat{K}}_{V}=9\cdot R\cdot \frac{T^{2}}{\Theta_{D}^{3}}\cdot \int_{0}^{\frac{\Theta_{D}}{T}}\frac{x^{4}}{{\left(e^{x}-1\right)}^{2}}\;dx$$

In Figure 7a, the molar isochoric entropy capacity according to Equation (Equation 15) is plotted as a function temperature for five different Debye temperatures. The examples were taken from Debye’s original work [11] and correspond to extrapolations for lead ($\Theta_{D}=95\;K$), silver ($\Theta_{D}=215\;K$), copper ($\Theta_{D}=309\;K$), aluminium ($\Theta_{D}=396\;K$) and diamond ($\Theta_{D}=1830\;K$, accurate description is given in Figure A2). The parabolic low-temperature course according to the phonon-related part of Equation (Equation 17) is also shown for each Debye temperature. The corresponding graphs for the molar temperature coefficient of energy ${\hat{C}}_{V}$ versus temperature are plotted in Figure 7b, which follows from combining Equations (Equation 4) and (Equation 15).

Remember that the purpose of considering the “heat capacity” $C_{V}$ is to estimate the amount of entropy stored [3] (p. 32, 353f), which can easily be estimated by visually integrating the graphs in Figure 7a where the entropy stored in 1 mol lead at 300 K is many times over the entropy stored in 1 mol diamond at the same temperature, which is not obvious from Figure 7b.

## 11. Phononic and Electronic Contributions to Entropy Capacity

In the low-temperature limit of the Debye model, the electronic and phononic contributions to the molar temperature coefficient of energy ${\hat{C}}_{V}$ are traditionally considered according to Equation (Equation 16) [36]:(16)$${\hat{C}}_{V}=\gamma \cdot T+\beta \cdot T^{3},$$

Combining Equation (Equation 16) with Equation (Equation 4), the electronic and phononic contributions to the molar entropy capacity ${\hat{K}}_{V}$ in the low-temperature limit of the Debye model are obtained:(17)$${\hat{K}}_{V}=\gamma +\beta \cdot T^{2},$$

Here, γ is the molar isochoric entropy capacity of the electron gas, often called the Sommerfeld coefficient [36] (p. 47):(18)$$\gamma =\frac{\pi^{2}}{3}\cdot R\cdot k_{B}\cdot D\left(E_{F}\right),$$

Here, $k_{B}$ is Boltzmann’s constant and $D\left(E_{F}\right)$ is the electronic density of states at the Fermi energy $E_{F}$.

The coefficient β in the phononic contribution to the entropy capacity is as follows [36] (p. 459) and allows us to estimate the Debye temperature $\Theta_{D}$:(19)$$\beta =\frac{12}{5}\cdot \pi^{4}\cdot R\cdot {\Theta_{D}}^{-3}$$

Equation (Equation 17) gives a physical meaning to the so-called Sommerfeld coefficient γ, which is the electronic contribution to the entropy capacity. Obviously, the electronic contribution to the entropy capacity is independent of temperature in the low-temperature approximation of the Sommerfeld–Drude model.

To retrieve the coefficients γ and β in Equation (Equation 16), ${\hat{C}}_{V}\;/\;T$ is often plotted as a function of $T^{2}$ without reference to the entropy capacity. However, as mentioned previously, it is indeed the molar entropy capacity ${\hat{K}}_{V}$, which is plotted versus $T^{2}$. Examples of ${\hat{K}}_{V}$ regression lines to empirical ${\hat{K}}_{p}$ data according to Equation (Equation 17) are shown in Figure 8 for gold, silver, copper and aluminium in the nonsuperconducting state. Note that ${\hat{K}}_{V}$ (model) and ${\hat{K}}_{p}$ (empirical) are considered to coincide at very low temperatures.

It is obvious from Table 2 that values for the Debye temperature in [55,56] (Figure 8) differ from the values in [11] (Figure 7). The former values are higher because they were obtained from fitting to the empirical data obtained at lower temperatures compared to the latter.

## 12. Discussion

### 12.1. Thermal Capacity

The storable thermal quantity is not the energy form “heat”, but the fluid-like quantity entropy. Thus, it is reasonable to associate the term thermal capacity with entropy capacity, which has been widely used implicitly. Its explicit use comes with the advantage of a descriptive fundamental understanding of thermal processes. Entropy capacity is a susceptibility and its inverse relates the response of the solid (change of temperature) to a stimulus (change of entropy contained).

### 12.2. Units of Entropy and Entropy Capacity

To emphasise that entropy is a countable quantity in its own rights, Wiberg [10] introduced for its units the special name *Clausius* ($1\;\mathrm{Clausius}=1\;\mathrm{cal}\cdot K^{-1}=4.1868\;J\cdot K^{-1}$). The use of the special name *Carnot* for the unit of entropy ($1\;\mathrm{Carnot}=1\;\mathrm{Ct}=1\;J\cdot K^{-1}$ [25,57]), which goes back to a proposal by Callendar [15], has also been suggested. With respect to Figure 2a, it is then possible to state that 1 mol graphite contains an amount of entropy of 5.66Ct at 300 K and 5.66Ct+8.95Ct=14.61Ct at 600 K, which sounds better than saying $5.66\;J\cdot K^{-1}$ or $14.61\;J\cdot K^{-1}$. Following this approach, the entropy capacity of 1 mol graphite at 600 K can be expressed as $0.029\;\mathrm{Ct}\cdot K^{-1}$, i.e., Carnot per Kelvin, which sounds better than $0.029\;J\cdot K^{-2}$.

It is a curious irony that a quantity that is central not only to thermal processes but involved in all dissipative (i.e., irreversible) physical processes has not yet received a special name in the International System of Units (SI).

### 12.3. Confusion and Resolution

Zemansky [2] (p. 76) summarised that the “idea” of heat (in older theories) as a form of energy was put forward in 1839 by Séguin and in 1842 by Mayer. Experiments by Joule during the period from 1840 to 1849 convinced the world. In 1847, von Helmholtz wrote a paper in which he applied Joule’s ideas to the sciences of physical chemistry and physiology. Fuchs [27] (p. 295) put forth the question “What did early experiments on heat prove?”. The short answer is that these “measurements were too crude” and “did not substantially add to the progress of thermodynamics”. Identifying heat with energy or an energy form was guided by prejudice rather than by a logical chain of reasoning.

“Heat capacity” has become a dead metaphor due to semantic shifts in the meaning of caloric (heat) during the development of thermodynamics from 1830 to 1850 [17,31,58]. Further dead metaphors are “heat storage”, “heat storage density”, “thermal energy storage density”, “heat reservoir” and “heat sink”, which generate images in mind that are inconsistent with thermodynamics. More examples are given in [31].

Semantic and conceptual impositions of the traditional mechanical theory of “heat” can be avoided if instead of thermal energy, entropy is seen as a resurrection of Carnot’s caloric (heat). This view follows notes by Ostwald (Ref. [59] p. 77, Ref. [17] p. 10, [60]) and others [15,16,27,58,61]. In the first edition of his famous book, Fuchs [27] (p. 289ff) clearly outlined the misconceptions of the traditional mechanical theory of “heat”, but reconciled it with the caloric theory of heat by identifying corresponding terms and definitions of both approaches.

To overcome the dichotomy between theory and clarity, several authors have suggested the correction of the semantics in thermodynamics [17,23,27,29,31,58]. The traditional “heat” should be substituted by thermal energy and entropy substituted by heat. The quantity entropy, which is mostly considered difficult, would become such a simple thing that “any school boy [and school girl] can understand” [15] and that “can be learned intuitively” [62].

## Figures and Tables

**Figure 1 entropy-24-00479-f001:**
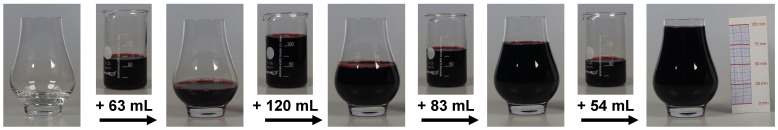
The capacity of a glass to store a fluid (here red wine) depends on the shape of the glass and changes with the fluid level. Depending on the shape of the glass vessel, different amounts of fluid are needed to raise the fluid level by 25 mm each. From left to right, the beakers contain 63 mL, 120 mL, 83 mL and 54 mL of fluid, which add to 320 mL when filled into the glass. A video sequence of filling the glass is available as Appendix A.

**Figure 2 entropy-24-00479-f002:**
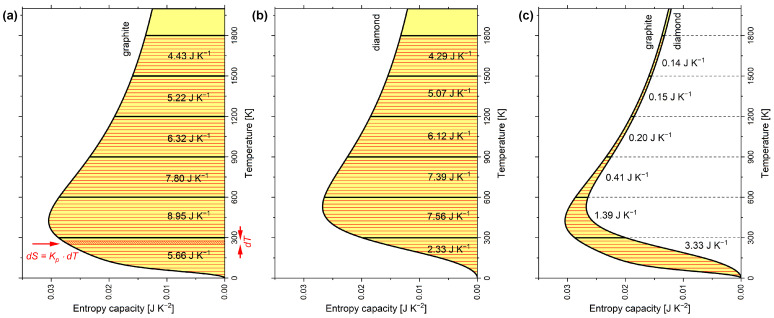
Temperature dependence of isobaric entropy capacity $K_{p}$ of 1 mol carbon allotropes: (**a**) graphite; (**b**) diamond; (**c**) graphite and diamond with differences highlighted. Entropy capacities were calculated according to the multiparameter model by Vassiliev and Taldrik [35] (see Section A.3). Following Wiberg [10].

**Figure 3 entropy-24-00479-f003:**
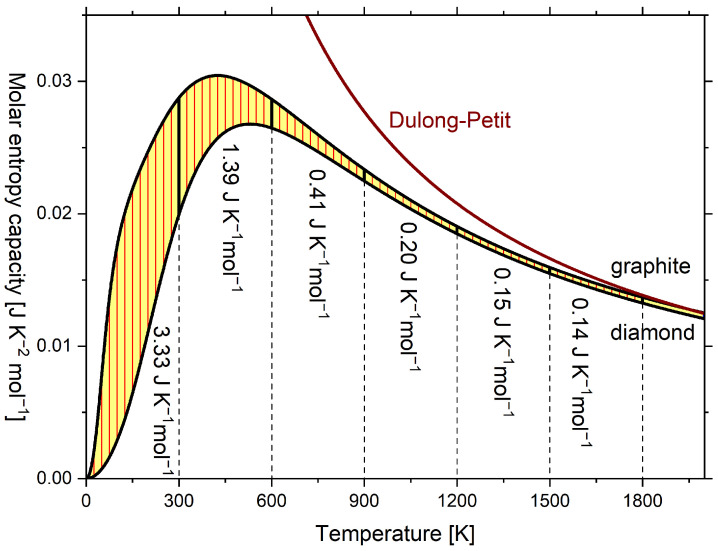
Temperature dependence of molar isobaric entropy capacity ${\hat{K}}_{p}$ of graphite and diamond and temperature dependence of molar isochoric entropy capacity ${\hat{K}}_{V}$ according to the Dulong–Petit relationship (hyperbolic) of the classical ideal gas. Isobaric entropy capacities were calculated according to the multiparameter model by Vassiliev and Taldrik [35] (see Section A.3). Following Wiberg [10].

**Figure 4 entropy-24-00479-f004:**
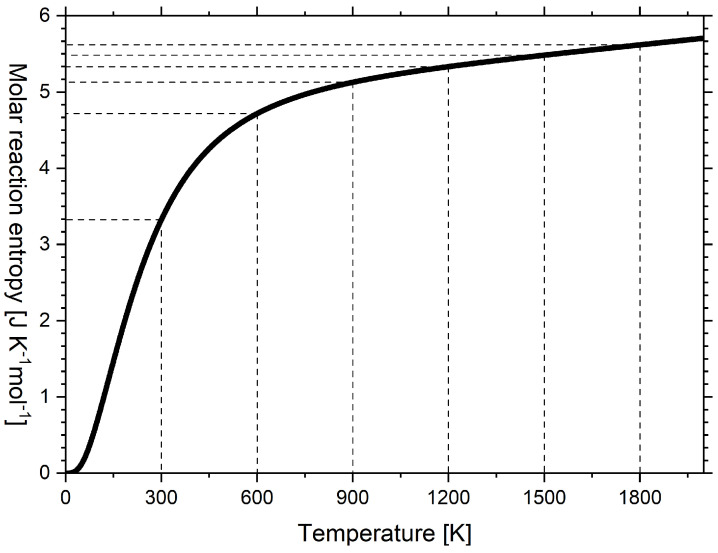
Molar reaction entropy of the transformation of diamond into graphite versus temperature as calculated according to the multiparameter model by Vassiliev and Taldrik [35] (see Section A.3). Following Wiberg [10].

**Figure 5 entropy-24-00479-f005:**
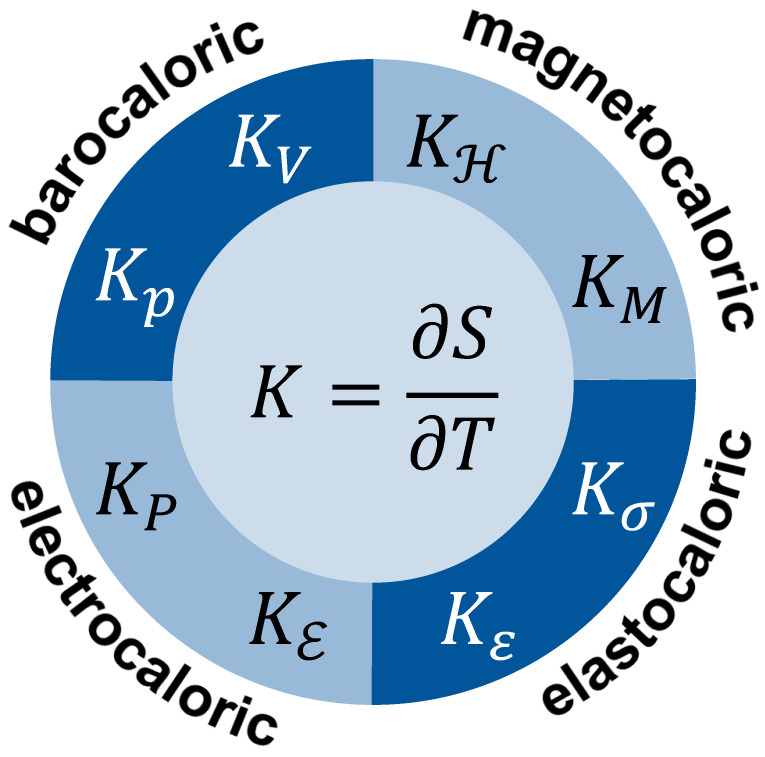
Examples of entropy capacity *K* of caloric materials at different intensive or extensive quantities being constant.

**Figure 6 entropy-24-00479-f006:**
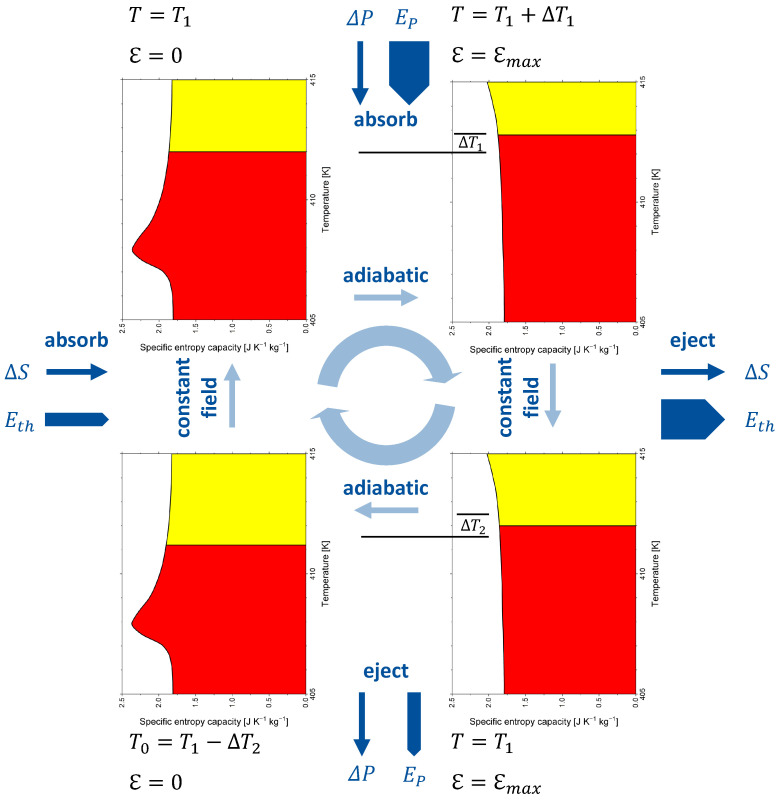
An electrocaloric cycle of barium titanate interpreted using entropy capacity. Irreversibility is omitted for clarity. A video sequence of squeezing and relaxing a fluid-filled vessel is available as Appendix A as an analogon. Following Scott [48].

**Figure 7 entropy-24-00479-f007:**
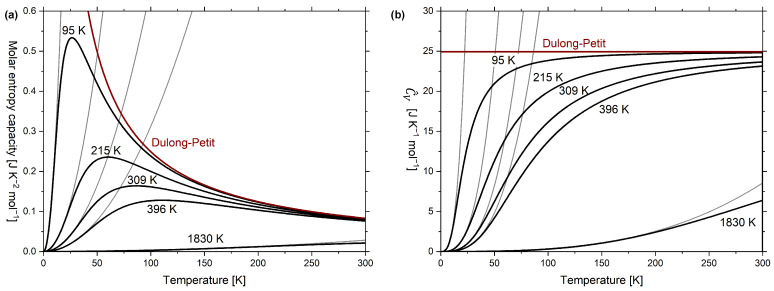
(**a**) Graph of isochoric molar entropy capacity versus absolute temperature; (**b**) graph of molar temperature coefficient of energy ${\hat{C}}_{V}$ versus absolute temperature. The graphs were calculated according to the Debye model for five different Debye temperatures on the examples given in Debye’s original work [11] and include low-temperature approximations (i.e., $T^{2}$ dependence in (**a**) and $T^{3}$ dependence in (**b**)).

**Figure 8 entropy-24-00479-f008:**
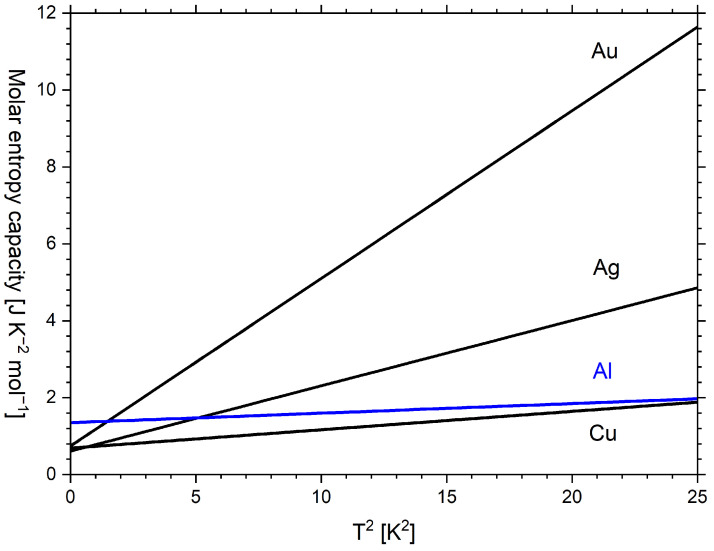
Plot of regression lines to the molar entropy capacity versus temperature squared. Based on parameters from [55] (Au, Ag, Cu) and [56] (Al) as given in Table 2.

**Table 1 entropy-24-00479-t001:** Energy forms in the context of caloric effects and related intensive and extensive quantities.

Caloric Effect	Energy Form	Conjugated Quantities			
		**Intensive Quantity**		**Extensive Quantity**	
magnetocaloric	magnetisation energy	magnetic field	H	magnetisation	*M*
elastocaloric	elastic energy	stress	σ	strain	ε
electrocaloric	polarisation energy	electrical field	ℇ	polarisation	*P*
barocaloric	compression energy	pressure	*p*	volume	*V*
all	thermal energy ^1^	temperature	*T*	entropy	*S*

^1^ Thermal energy is also called “heat”.

**Table 2 entropy-24-00479-t002:** Comparison of the molar electronic entropy capacity γ (Sommerfeld coefficient) and Debye temperature $\Theta_{D}$ for the elements displayed in Figure 7 and Figure 8.

Substance	γ	$\Theta_{D}$	
	(mJ K${}^{-2}$ mol${}^{-1}$)	(K)	(K)
Au	0.743 [55] ^1^	164.57 [55] ^1^	N/A [11]
Ag	0.610 [55] ^1^	225.3 [55] ^1^	215 [11]
Cu	0.688 [55] ^1^	343.8 [55] ^1^	309 [11]
Al	1.35 [56] ^2^	427.7 [56] ^2^	396 [11]

^1^ Data from [55] refer to a corrected 1948 helium vapor pressure–temperature scale. ^2^ Data from [56] refer to the
1959 helium vapor pressure–temperature scale.

## Data Availability

Not applicable.

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
