# Peer review of "On the Thermal Capacity of Solids"

_entropy, 2022, doi:10.3390/e24040479_

Round 1

Reviewer 1 Report

This is a well written and timely study of an important subject in thermodynamics that is usually neglected. The author shows that, even if entropy capacity is not mentioned directly in most work in thermodynamics, it exists implicitly and, if used explicitly, leads to substantial gains for anyone trying to understand thermal science and applications.

I only have two or three concrete recommendations:

I would reverse the logical order in sentences describing the relation between changes of entropy and changes of temperature (such as on lines 5-7) to make them conform to quotes of Wiberg's work (starting on line 122).

In diagrams showing molar entropy capacity, I would do this explicitly, i.e., change the name and the units on the pertinent axis.

The term for traditional thermodynamics is "mechanical theory of heat" rather than "kinetic theory of heat." (Paragraph starting on line 440.)

A question: Is Appendix C missing? In the file I received, there is a title but no content.

English language needs to be polished lightly. 

Reviewer 2 Report

Very interesting work dealing with aspects of naming errors of certain sizes. I am asking the author to pay attention to a few small elements:
- the SI unit of liter is a lowercase letter l and not a capital letter, as e.g. the author uses 1 mL in the figure instead of ml, and similarly in the content;
- in line 28 and 88, for example, "Falk & Ruppel", instead of '&' there should probably be 'and' according to the journal's requirements;
- in references, please separate Notes from literature - here should be only literature, not e.g. items 1,6,9,14,29,31,32 - please reorganize it according to the journal's requirements.

Reviewer 3 Report

The work tries to address the confusion regarding entropy, entropy capacity and thermal capacity. These terms are fundamental in thermodynamics and yet poorly understood, and hence the presented work is important and appreciated. There are minor grammatical problems that could be easily fixed. Example: 

Line 174: The capacity of a glass to store a fluid depends its shape

that should read

The capacity of a glass to store a fluid depends on its shape

I recommend acceptance of the paper after addressing these minor grammatical issues.
